# Household Dietary Diversity among the Ethnic Minority Groups in the Mekong Delta: Evidence for the Development of Public Health and Nutrition Policy in Vietnam

**DOI:** 10.3390/ijerph20020932

**Published:** 2023-01-04

**Authors:** Hiep N. Le, Kien V. Nguyen, Hai Phung, Ngan T. D. Hoang, Duong T. Tran, Lillian Mwanri

**Affiliations:** 1Department of Food Technology, An Giang University, Long Xuyên 880000, Vietnam; 2Center for Educational Testing and Quality Assessment, Vietnam National University HCMC, Ho Chi Minh City 700000, Vietnam; 3School of Medicine and Dentistry, Griffith University, Gold Coast, QLD 4222, Australia; 4Health & Agricultural Policy Research Institute, University of Economics Ho Chi Minh City, Ho Chi Minh City 72516, Vietnam; 5Climate Change Institute, An Giang University, Long Xuyên 880000, Vietnam; 6Research School of Management, College of Business and Economics, The Australian National University, 26 Kingsley St., Acton, Canberra, ACT 2601, Australia; 7National Institute of Nutrition, 48B Tang Bat Ho, Pham Dinh Ho Ward, Hai Ba Trung District, Hanoi 100000, Vietnam; 8Research Centre for Public Health, Equity and Human Flourishing, Torrens University Australia, Adelaide Campus, Adelaide, SA 5000, Australia

**Keywords:** dietary diversity, ethnic minority populations, nutrition policy, Vietnam

## Abstract

Poor household dietary diversity has been linked to malnutrition in individuals, households, and cumulatively in populations. High rates of malnutrition among Khmer ethnic children aged five years and younger have been reported in Tri Ton district, Vietnam. This paper aims to further investigate household dietary diversity and associated factors among Khmer ethnic minority populations in Vietnam. A cross sectional study was conducted from October 2018 to April 2019 in Tri Ton District, An Giang Province. By employing a multistage sampling technique, a total of 402 (99.8% response rate) participants were interviewed to measure household dietary diversity using a structured and validated questionnaire developed by FAO. Both bivariate and multivariate logistic regression analyses were carried out to identify factors associated with household dietary diversity. The results showed that the prevalence of low, medium and high dietary diversity scores were 21.4%, 70.4% and 8.2%, respectively. Male-headed households, literacy level, household income, exposure to mass media on nutrition and health information, and frequency of eating were positively associated with household dietary diversity (*p* < 0.05). However, owning a vegetable and rice farm was not statistically related to households’ dietary diversification. The paper concludes that the magnitude of household diversified dietary intakes was essentially low to medium in participants’ households. These findings have provided evidence to inform the development of the National Nutrition Strategy—2021–2030 in Vietnam, to be revised in 2045. This national strategy proposes appropriate interventions, programs and policies to improve socioeconomic status in ethnic groups and in mountainous areas to enhance populations’ health and well-being including controlling childhood malnutrition. In order to improve population health and wellbeing in Tri Ton District, further actions to address effective dietary practices including strengthening nutrition and health communication about the need to improve household dietary diversity to high levels are recommended.

## 1. Introduction

Malnutrition is a serious and significant public health problem linked to high mortality and morbidity risks, particularly among children aged five years or younger [1,2] (from here on to be termed as under-fives). Globally in 2017, a significant number of malnourished under-fives, including 150 million, 50 million and 38 million were reported to suffer from stunting, wasting and overweight, respectively [3]. Most of these children were from: (i) poor rural areas of low- and middle-income countries [4,5], (ii) ethnic minority communities, and (iii) indigenous people [6,7]. According to previous studies, malnutrition accounts for more than 3.1 million deaths per year in under-fives, with poor dietary diversity being one of the key and modifiable determinants [8,9,10]. Dietary diversity refers to the consumption of food, across and within food groups, capable of ensuring sufficient intake of essential nutrients that can promote physical and mental health and wellbeing [11]. A growing body of literature has demonstrated a significant association between dietary diversity, micronutrient adequacy and positive health outcomes [12,13,14]. It has also been noted that when more food groups are included in daily diets, the likelihood of fulfilling the nutrient requirements increases [15]. By contrast, a low dietary diversity: increases the risk of malnutrition including underweight and stunting [16], triggers cognitive deficits [17], and raises the proportion of undernourished populations [18]. The socio-economic disparity has also been noted as one of the explanatory factors for under-nutrition and poor health outcomes in populations [19,20,21]. It is acknowledged that people with a high socio-economic status (SES) are more likely to have resources enabling them to have good dietary diversity, whereas people with low SES are more likely to have poor dietary habits leading to poorer health outcomes [22]. In addition to socioeconomic differences, ethnic minority groups are at high-risks in terms of household dietary diversity relative to the dominant ethnic groups due to the impact of their own culture, religious beliefs and rurarity [23].

In Vietnam, despite the recent great achievements in reducing poverty and hunger, there is still a disparity in health outcomes with ethnic minority groups in rural and remote areas continuing to suffer from significant malnutrition [24,25]. Several policies and programs, along with significant investments, have been allocated to vulnerable communities such as the Khmer minority group to improve livelihood, health and nutrition status [26]. The recent National Nutrition Strategy and the targeted national program for socio-economic development in ethnic minorities and mountainous areas, have also created a supportive environment for the comprehensive development of ethnic minority populations [27,28]. The Khmer group, who primarily live in the Mekong Delta, face significant challenges such as: low quality of education, poor socioeconomic status, job uncertainties and slow adaptation to the business environment [29]. These challenges have led to limited average income per person compared with the national level (VND 1.626 million per month versus 4.16 million per month in 2016) [30,31], leading to poor purchase power, thus preventing the consumption of HDD. Low literacy levels and language barriers are also among the challenges for Khmer people to integrate into the Vietnamese mainstream education system—leading to low health and dietary literacy [32,33]. The level of malnutrition among Khmer under-fives in Tri Ton District of the Mekong Delta was higher at 37.8% for underweight (WAZ < −2), 50.0% for stunting (HAZ < −2) and 17.0% for wasting (WHZ < −2) compared with the national average of 14.1% underweight, 24.6 stunting and 6.4% wasting, respectively [32,34]. The Tri Ton district is one of the poorest communities in An Giang province. The most recent data showed that people in 14/15 communes in the district completed the primary school literacy [35]. The livelihood of local people depends heavily on forest and small-scale cultivation. Meanwhile, the district has been affected by climate change, especially drought caused by extreme weather events [36]. It is widely accepted that climate change has an impact on all aspects of food security and nutrition including food production, food accessibility, food utilisation, and dietary diversity, but more severe consequences on minority groups such as Khmer than on the Kinh group (account for 85% of the Vietnamese population) [37,38]. A wide range of international policies have been suggested for nations to reduce levels of undernutrition in under-fives. Poverty and malnutrition reduction were the Millennium Development Goals (MDGs) goal [39] by 2015. However, the situation of undernutrition of rural poor households, especially Khmer households in the Tri Ton district, a rural district of An Giang province remained high at the conclusions of MDGs in 2015 [32]. While food security is considered as one of the determinants of childhood malnutrition [40], and have been targeted in nutrition policy in Vietnam, household dietary diversity has not been specifically focused. Additionally, our literature review showed a limitation of evidence for characteristics of household dietary diversity in Khmer minority group. This study was conducted to assess Khmer’s household dietary diversity and associated factors in Tri Ton District, An Giang Province, Mekong Delta Vietnam. Since the study was conducted, the findings have already provided evidence for the 2021 to 2030 National Nutrition Strategy. The strategy is envisioned to be revised in 2045 and its targets are to enhance ethnic minority populations’ health and well-being and to control childhood malnutrition.

## 2. Methods

### 2.1. Study Design, Settings and Participants

A cross-sectional community-based study was conducted from October 2018 to April 2019 in Tri Ton district, a rural district of An Giang Province in Mekong Delta region, Vietnam. (Figure 1). The Tri Ton District is located in the South-western part of the country. In 2018, the district had a total of 33,674 households, of which Khmer ethnic minority households accounted for 33.4% (11,263 households) [33]. In recent years, the number of Khmer migrants seeking employment in large cities and industrial parks has increased, with the young working age populations being over represented [41].

### 2.2. Sample Size and Sampling Technique

The sample size for this survey was determined by the single population proportion formula [43].
$$n=Z^{2}(1-\alpha /2)\frac{p(1-p)}{d^{2}}$$
with *p*-value = 50%, without any assumption prevalence in this area, 95% confidence interval = 1.96; *d* = 0.05, *z* = the standard normal tabulated value, and α = level of significance. Therefore, the total sample size was 384. Considering 5% of the non-response rate, the total sample size was 403.

The multistage sampling technique was employed to select the study participants in Tri Ton district. Firstly, of nine communes where the Khmer live, three communes were selected using simple random sampling method. Secondly, the proportional to population size allocation (PPS) was conducted to calculate the sample size for each commune. Thirdly, simple random sampling method (conducted by choosing a random number in the random number table corresponding to a numbered list of Khmer households) was used to select the households for interviews.

### 2.3. Data Collection Procedures

A house-to-house approach was carried out for data collection. A standard questionnaire in English was translated into Vietnamese. Because not all Khmer people speak Vietnamese, interpreters were used during the interviews with participants. The translated question was tested using a sample of 20 households (5%) (not included in the study sample) in a community which had similar characteristics of the research sites. Before the actual data collection, the questionnaire was revised based on their comments prior to pilot testing and validation for internal consistency. This process was necessary to ensure that the translated questionnaire was similar to the original English questionnaire. For data collection, the person who was responsible for preparing household food in the previous day was interviewed on behalf of household members for 24 h food recall and SES data.

Unattended households were revisited twice during the visit day to ensure that the required sample size was reached. Six research trained students and three local Khmer public health practitioners (interpreters) were involved in the data collection process. Training of interviewers and field work supervision was carried out by the principal investigator. The data were immediately checked in the field for accuracy and completeness. Additionally, supervisors re-checked the questionnaire for the completeness, quality, and consistency of information collected on the daily basis.

### 2.4. Measurement of Study Variables

A standard multi-part questionnaire was used to assess household dietary diversity as an outcome variable and some other explanatory variables.

The dependent variable (household dietary diversity) was collected using guidelines for measuring household dietary diversity developed by FAO [44]. Dietary diversity score (DDS) was defined as the number of different food groups consumed by family members over the last 24 h [44]. One point was awarded to each food group consumed over the reference period, and the sums of all points were calculated for the DDS for each household [44]. Low Dietary Diversity, Medium Dietary Diversity, and High Dietary Diversity were defined as less than three, 4–6, and ≥7 food groups consumed, respectively [45,46].

Independent variables included: demographic and socioeconomic factors such as sex, age, education level of the head of the household (Vietnamese), main occupation of the head of household, household income (million Vietnam Dong (VND)/month), media exposure, vegetable farm owner and rice farm ownership.

### 2.5. Data Management and Statistical Analysis

The data was checked for completeness and consistency before cleaning and entry intoEPI-INFO version 7 statistical software [47]. Data analyses were performed by SPSS software package version 20 [48].

Age was categorised based on the encyclopaedia of Aging and Public Health [49]. Education level was classified as illiteracy and literacy in Vietnamese. Average household income was classified as ≤2.0, 2.1–4.0, 4.1–6.0, and >6.0 million Vietnamese Dong/month [50]. Medium Dietary Diversity and High Dietary Diversity were categorised as Diversified category, Low Dietary Diversity as Non-diversified category [45,51].

Descriptive statistics were performed to characterise the participants using different variables of interest. The association between household dietary diversity and socio-demographic, economic, and dietary variety variables of respondents was assessed using univariate logistic regression followed by a multivariate forward stepwise logistic regression test to determine the adjusted odd ratio (aOR). Hosmer and Lemeshow goodness-of-fit test was performed. The area under the receiver-operating characteristic (ROC) curves was calculated to evaluate the predictive value of various factors. The ROC curve assesses the capacity of the entire logistic regression model to distinguish between Khmer households with diverse and non-diverse diets. Results of a test are deemed statistically significant if the *p*-value is less than 0.05.

## 3. Ethical Consideration

Ethical clearance was obtained from the Ethical Review Committee by The board of Rectors of An Giang University (now it is An Giang University—Vietnam University Ho Chi Minh City) on 1 October 2018 before the study began. There was also permission obtained from the authority and administration health officers of 3 communes in Tri Ton district, Vietnam who allowed and supported the researchers to conduct the research in the district. Each participant’s informed consent was obtained after the researchers explained the purpose of the study. Participants were informed of the voluntary nature and their right to refuse participation at any stage of the study. They were also assured of the confidentiality of the collected information.

## 4. Results

### 4.1. Characteristics of the Participated Households and Household Dietary Diversity

Of the 403 eligible respondents, 402 participated in this study, a response rate of 99.8%. As presented in Table 1, the proportion of female household heads, illiteracy (Vietnamese), and age range from 41 to 65 years were 40.3%, 21.9% and 54.5%, respectively. More than 60% of the households had average monthly income between 2.1 and 4.0 million VND. Over a third of households in this sample had a vegetable or rice farm. The mean household dietary diversity score was 4.6 (SD ± 1.4). The majority of the households were at the moderate level of dietary diversity (over 70%). The proportion of households classified as having diversified dietary intake was roughly 80%. Just over a third of households ate three times per day, with the majority of them eating two times a day. Meanwhile, nearly a half of households skipped breakfast.

The prevalence of participants who consumed four food groups accounted for the highest followed by 5, 6, and 7 food groups (Figure 2). The most consumed food groups were: cereals (100%); vegetables (68.7%); fish and seafood (67.9%); and spices, condiments and beverages (54.7%) (Figure 3). Meat/poultry/organ meats were consumed only by 35.1% of households. Fruits, eggs, dairy, nuts and root crops were consumed the least (less than 20%) (Figure 3).

### 4.2. Associated Factors of Household Dietary Diversity

Table 2 shows that sex and educational level of the household head, age, household income, media exposure on nutrition and health, frequency of eating, and breakfast eaten were significantly associated with diversified diets among Khmer households. Ownership of a vegetable or rice farm was not significantly associated with household dietary diversity.

When seven variables from the unadjusted model were included in a single model adjusting for each other, all factors remained significant except for age and breakfast eaten. The Hosmer and Lemeshow goodness-of-fit test showed the model being a good fit with *p* = 0.908. The multivariate logistic regression analysis showed significantly higher odds of having diversified diets among the households with male-headed families (aOR 3.34; 95% CI 1.86–5.98, *p* = 0.000), literacy (Vietnamese) (aOR 9.32; 95% CI 5.06–17.16, *p* = 0.000), household income (million VND/month) 2.1–4.0 (aOR 2.87; 95% CI 1.56–5.29, *p* = 0.001), 4.1–6.0 (aOR 3.64; 95% CI 1.03–12.93, *p* = 0.046), >6.0 (aOR 6.27; 95% CI 1.09–36.06, *p* = 0.040), having media exposure on nutrition and health (aOR 1.89; 95% CI 1.04–3.41, *p* = 0.036), and frequently ate three times and above per 24 h (aOR 2.12; 95% CI 1.13–3.98, *p* = 0.019) (Table 3).

A ROC curve was developed (Figure 4) from the fitted predicted value from the final logistic regression model. The area under the operator characteristic curve was 0.835.

## 5. Discussion

### 5.1. Household Dietary Diversity among Khmer Households

Although the household dietary diversity score (HDDS) does not measure dietary diversity consumed at an individual level, it does indicate household eating patterns of a range of foods, information that can be used to promote good eating patterns within households and communities for prevention of, not only undernutrition, but over nutrition including obesity [53]. The HDDS has also been highly recommended as essential for modifying the risks of several dietary-related chronic diseases [54]. The mean HDDS of this study was consistent with the previous studies in Cambodia [55], where 97.6% of the population is ethnically Khmer, and greater than 80% completed primary school education [56]. In contrast, the mean HDDS from our study was lower the 2011, survey results from four provinces of non-Khmer group [57]. A number of studies in low- and middle-income nations also found similar results where most ethnic groups had low or medium dietary diversity scores [45,58,59,60]. Thus, ethnicity may have an impact on HDDS as people in the same ethnic group are likely to have similar dietary patterns and eating behaviours independent of socioeconomic status. However, this study did not explore this issue, and further studies are recommended to in-depth investigate this statement. In addition, a considerable proportion of households at Low Dietary Diversity (consumed less than three food groups per day) in this study may result in increased malnutrition in the households [16,61]. The future studies regarding to HDD should consider ethnicity in order to control malnutrition.

Results from the general nutrition survey in Vietnam in 2019 revealed that the Vietnamese are consuming more meat, vegetables but less starchy food [62]. In contrast, the dietary pattern of the studied ethnic group seemed to be characterised by starchy and vegetable-based foods. Based on the socioeconomic status of these populations, the plausible reasons for these dietary patterns could be that Khmer households were mainly surrounded by the natural forest [33], which would have made it easier to harvest wild vegetables for their meals. Moreover, while young Khmer people were in the cities and industrial zones [41], the older remained at home to take care of the children and preparing meals for the family. The traditional custom and eating habits from the older generations which persist in contemporary households, is the predominant consumption of starchy and vegetables food-based diet. It is reasonable to argue that the Khmer households had similar dietary patterns to that of ethnic minorities in Northwest Vietnam where grains, grain products and vegetables have been reported to be the most frequently consumed foods [63], and where fruits and legumes, nuts and seeds are rarely consumed. These findings are consistent with Nguyen and colleagues’ findings, where starchy staple foods and vegetables were reported to be widely consumed in other low- and middle-income countries including Bangladesh, Ethiopia, Mexico, and Botswana. In these countries, meat, eggs, dairy products, and fruits were reported to be less likely to be consumed in the previous 24 h [64,65,66]. It is therefore reasonable to allude that the current dietary consumption of the Khmer minority households comprises a monotonous pattern of starchy based staples, with inadequate animal products and fruits, potentially lacking essential micronutrients for the human development. As such, nutrient deficiencies and malnutrition patterns in many of the households were similar to commonly found some communities in developing countries where foods consumption is consistent with having poor dietary patterns [67,68]. While we acknowledge that practicing dietary diversity is important, it is worth noting that improved dietary diversity can be challenging due to a range of factors including, socio-economic, residence and cultural factors [69,70], especially where traditional food menu with low diversity are favoured. We also must acknowledge that, the differences in HDDS might be due to variations in agroecology, culture and guidelines used to assess the dependent variables. As these factors were not the focus of the current study, further studies are recommended to explore the role that these factors may play in contributing or not to HDDS.

### 5.2. Associated Factors of Household Dietary Diversity among Khmer Ethnic Group

Understanding the factors contributing to household dietary diversity helps to develop appropriate food security policies and interventions. Consistent with the literature review for general populations, our final logistic regression model revealed that factors such as the sex and educational level of the household head, household income, media exposure to nutrition and health, and frequency of eating were significantly associated with diversified diets in the Khmer group. This model had an excellent predictive accuracy for diversified diets among Khmer households, with the area under the ROC curve being 0.835 [71].

Higher odds for household dietary diversity among males compared to females being the household’s head were consistent with findings from studies on rural households in low- and middle-income countries or settings such as West Bengal, India [58], Sri Lanka [72], Mirab Abaya Woreda, Southern Ethiopia [46], Finote Selam town, Ethiopia [45], Amatole and Nyandeni districts, South Africa [59]. It is evident that females who were heads of households were also working. Therefore, increased women’s workload could have negatively impacted their nutritional intake and quality because of factors such as the reduction of time available for family nutrition and health-enhancing activities. This could possibly mean that females (who in many cultures are food makers) [73] did not have sufficient time and resources required to provide optimum dietary diversity for their households [74,75,76]. This is contrary to males as the household’s head. Additionally, the income of males may be higher than females [77], leading to high purchasing power including of diverse foodstuffs. However, this statement was not studied in our research and needs further investigation.

The level of literacy was another factor to have a positive significant association with household dietary diversity. This finding is significant, because it calls for efforts to promote education of communities in the quest to improve dietary diversity. Similar and comparable findings have been suggested by other studies [51,78,79,80], indicating that people with higher level of education have a greater awareness and understanding of nutritional and health benefits of high dietary diversity, and thus leading better dietary practices.

Household income was also positively statistically significantly associated with HDDS. Studies by Belachew and Yemane [51], Taruvinga et al. [59] and Morseth et al. [81], have also revealed direct association between higher income and higher diversified diet in households, which may indicate the need to address poverty to improve dietary diversity. These findings support the WHO social determinants of health framework where high education and higher wealth would address poverty and provide resources to ensure adequate nutritious food supply [82]. Several authors have also argued that the increased demand for fruits (which could mean better dietary quality) increases with income [83,84,85]. It is agreed that nutritious and healthy foods including fruits may be expensive, and low income households may not choose these as a priority due to the need to fulfil other pressing life demands [86].

Media exposure on nutrition and health was another factor that showed significant association with household dietary diversity. In conformity with findings by Beyene and colleagues [87], the odds of household dietary diversity were 1.89 times higher among households that were exposed to mass media on nutrition and health compared to the households unexposed to mass media. It is reasonable to hypothesis that this finding could be the impact of currently running media promotions that use radio and television to promote healthy eating and healthy cooking practices.

Consistent with the findings by Mekuria and colleagues [45], the odds of household dietary diversity were 2.12 times higher among households who consumed food three times and over, compared to households that consumed food two times. This is logical and could be explained by the fact that increasing the frequency of consuming food would likely increase the type of items consumed, and hence increased dietary diversity of the households.

There was, however, no statistically significant correlation between having a vegetable or rice farm and dietary diversification. This conclusion is consistent with research conducted in Finote Selam town, northwest Ethiopia [45], but dissimilar to studies conducted in Noakhali district, Bangladesh [88], Tanzania [89], and West Bengal, India [58]. According to the findings of these studies, the land is a crucial factor for diet variety, and families owning larger areas of agricultural land had higher HDDS. It has also been reported that dietary diversity increases with farm size [58,88,89]. In this research, however, the average acreage of vegetable farms and rice farms held by Khmer families was small. Therefore, there was no difference in dietary diversity between farm-owning and non-farm-owning households.

The study has already made significant contributions to the body of knowledge, including informing the development of the National Nutrition Strategy for the period of 2021–2030, envisioned to be revised in 2045. From our understanding, this is the first research on the Khmer minority group, particularly for household dietary diversity. Results from this study will additionally raise awareness among the community and policymakers regarding improving household dietary diversity to control undernutrition.

## 6. Conclusions

The prevalence of low, medium and high dietary diversity scores were 21.4%, 70.4% and 8.2%, respectively. Dietary patterns of Khmer ethnic groups were characterised by starchy, staple foods and vegetables with low frequency of animal protein source foods, dairy products, nuts, and fruits consumed. Sex of household head, education level of household head, household income, frequency of eating and exposure to mass media on nutrition and health were significant contributing factors for household dietary diversity. The government should enforce policies that ensure that diversified foods are available and accessible at low cost for the Khmer households and communities. Strengthening nutrition and health communication is recommended to improve Khmer household dietary diversity, resulting in improved nutrition and health of the Khmer communities. Findings from this research have provided evidence for the National Nutrition Strategies for the period of 2021–2030, envision to be revised in 2045. The Strategy proposes appropriate and effective interventions, programs and policies to improve socioeconomic status in ethnic groups and mountainous areas in Vietnam [27,28]. Prospective policy targeting childhood malnutrition control should be issued, and in consideration of diversity of culture, gender impact, appropriate education and communication channels. In order to improve population health and wellbeing in Tri Ton District, further efforts including strengthening nutrition and health communication about the need to improve household dietary diversity to high levels are recommended.

## 7. Limitations of the Study

The limitation of the current study is the reliance on 24 h dietary recall which does not show the usual dietary practices of individuals members in the household. It is well known that the dietary patterns of individuals or household change during religious festivals. To ensure that the religious festivals did not have impact on the current study findings, interviews were not conducted around the times of religious festivals including the traditional Tet Chol Chnam Thmay, Sen Don Ta Festival, Dang Y Kathinat festival, Ok Om Bok moon worship festival, Dang Bong festival, Buddha’s Birthday ceremony and the Phum Soc festivals. It is also worth noting that it did not compare the findings of the current study with the Kinh group in the district, which is a limitation to consider in interpreting the results. Given these limitations and the nature of the current study—a cross-sectional design, interpretations of conclusions must be undertaken cautiously.

## Figures and Tables

**Figure 1 ijerph-20-00932-f001:**
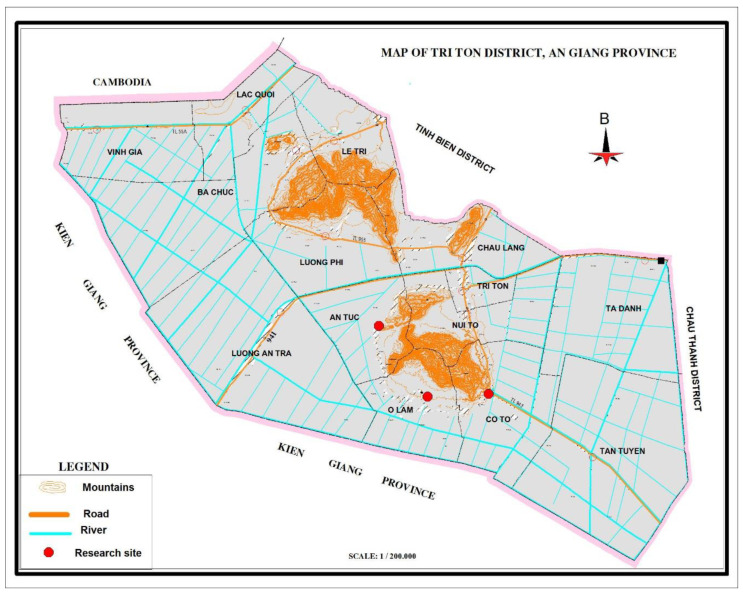
Map of Tri Ton district, An Giang province (Reprinted/adapted with permission from Ref. [42]. Copyright 2020, Tien Duy Pham).

**Figure 2 ijerph-20-00932-f002:**
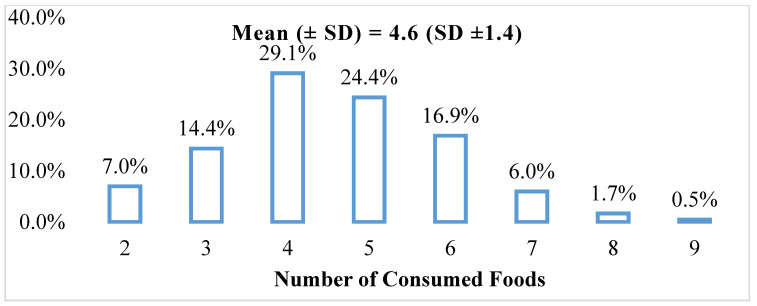
Food consumption pattern of Khmer minority households by the number of foods consumed (%).

**Figure 3 ijerph-20-00932-f003:**
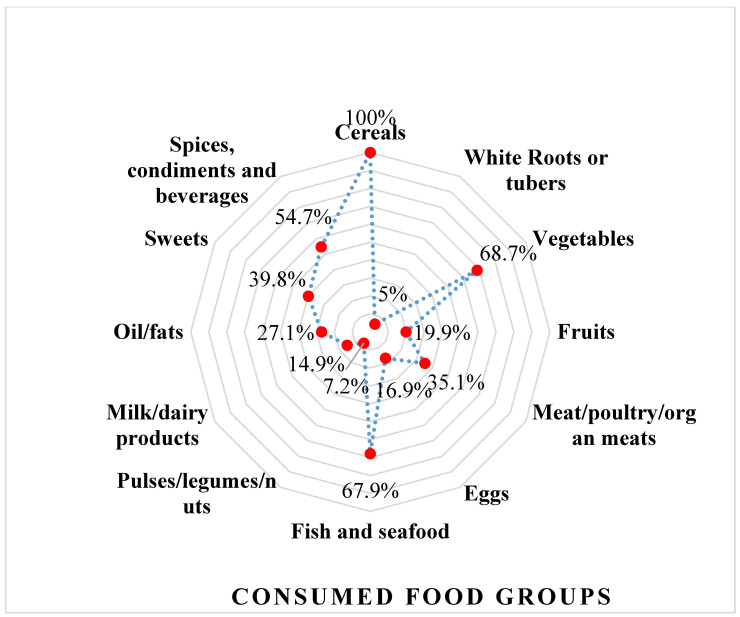
Food consumption patternsof Khmer minority households by consumed food groups (%).

**Figure 4 ijerph-20-00932-f004:**
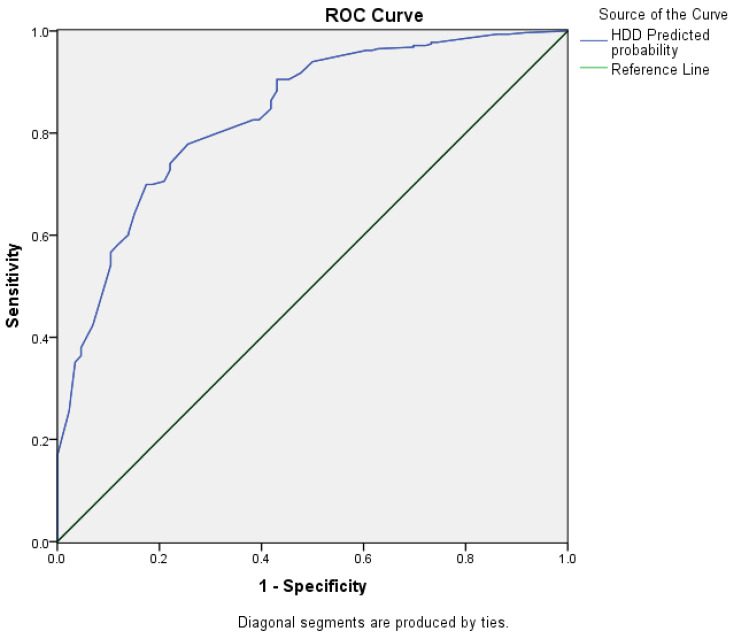
Receiver-operating characteristic curve for a multivariate logistic regression model to predict Khmer household dietary diversity in Tri Ton district.

**Table 1 ijerph-20-00932-t001:** Socio-demographic, economic, and dietary diversity characteristics of respondents (n = 402).

Variables		Frequency	Percent
**Socio-Demographic and Economic Characteristics**			
Research site	Co To	81	20.1
	An Tuc	80	19.9
	O Lam	241	60.0
Sex of household head	Male	240	59.7
Education level of household head	Illiteracy (Vietnamese)	88	21.9
Age of household head	>65	40	10.0
	41–65	219	54.5
	18–40	143	35.6
Main occupation of household head	Farmer	116	28.9
	Worker in company	7	1.7
	Daily laborer	136	33.8
	Small trader	64	15.9
	Housewife	74	18.4
	Government employed	5	1.2
Household income (million Vietnamese Dong/month) ^a^	≤2.0	99	24.6
	2.1–4.0	249	61.9
	4.1–6.0	34	8.5
	>6.0	20	5.0
Media exposure on nutrition and health ^b^	Yes	177	44.0
Vegetable farm owner	Yes	142	35.3
Rice farm owner	Yes	168	41.8
**Dietary diversity characteristics**			
Dietary diversity	Diversified	316	78.6
Dietary diversity level	Low	86	21.4
	Medium	283	70.4
	High	33	8.2
Frequency of eating (per day)	Two times	244	60.7
	Three times	154	38.3
	Four times and above	4	1.0
Breakfast eaten	Yes	224	55.7

^a^ 1 million Vietnamese Dong = 43 USD (Reprinted/adapted with permission from Ref. [52]. Copyright 2019, Joint Stock Commercial Bank for Foreign Trade of Vietnam (Vietcombank); ^b^ Media exposure on nutrition and health is defined as having exposure (at least one time in a week) with information regarding nutrition and health communication on the mass media.

**Table 2 ijerph-20-00932-t002:** Characteristics and univariate analysis of factors associated with Khmer household dietary diversity in Tri Ton district.

Variables	Dietary Diversity		OR (95% CI) ***	*p*-Value
	Non-Diversified * n (%)	Diversified ** n (%)		
Research site				
Co To ^†^	23 (28.4)	58 (71.6)	1	
An Tuc	14 (17.5)	66 (82.5)	1.87 (0.88–3.97)	0.103
O Lam	49 (20.3)	192 (79.7)	1.55 (0.87–2.76)	0.134
Sex of household head				
Female ^†^	53 (32.7)	109 (67.3)	1	
Male	33 (13.8)	207 (86.2)	3.05 (1.86–4.99)	0.000
Education level of household head				
Illiteracy (Vietnamese) ^†^	47 (53.4)	41 (46.6)	1	
Literacy (Vietnamese)	39 (12.4)	275 (87.6)	8.08 (4.73–13.82)	0.000
Age (years)				
65+ ^†^	17 (42.5)	23 (57.5)	1	
41–64	40 (18.3)	179 (81.7)	3.31 (1.62–6.76)	0.001
18–40	29 (20.3)	114 (79.7)	2.91 (1.38–6.14)	0.005
Main occupation of household head				
Non-farmer ^†^	68 (23.8)	218 (76.2)	1	
Farmer	18 (15.5)	98 (84.5)	1.70 (0.96–3.01)	0.069
Household income (million Vietnamese Dong/month)				
≤2.0 ^†^	40 (40.4)	59 (59.6)	1	
2.1–4.0	40 (16.1)	209 (83.9)	3.54 (2.10–5.99)	0.000
4.1–6.0	4 (11.8)	30 (88.2)	5.09 (1.66–15.55)	0.004
>6.0	2 (10.0)	18 (90.0)	6.10 (1.34–27.76)	0.019
Media exposure on nutrition and health				
No ^†^	58 (25.8)	167 (74.2)	1	
Yes	28 (15.8)	149 (84.2)	1.85 (1.12–3.05)	0.017
Vegetable farm owner				
No ^†^	58 (22.3)	202 (77.7)	1	
Yes	28 (19.7)	114 (80.3)	1.17 (0.71–1.94)	0.545
Rice farm owner				
No ^†^	52 (22.2)	182 (77.8)	1	
Yes	34 (20.2)	134 (79.8)	1.13 (0.69–1.83)	0.632
Frequency of eating (per day)				
Two times ^†^	65 (26.6)	179 (73.4)	1	
Three times and above	21 (13.3)	137 (86.7)	2.37 (1.38–4.07)	0.002
Breakfast eaten				
No ^†^	51 (28.7)	127 (71.3)	1	
Yes	35 (15.6)	189 (84.4)	2.17 (1.34–3.52)	0.002

* Non-diversified category was Low Dietary Diversity (≤3 food groups); ** Diversified category included Medium Dietary Diversity and High Dietary Diversity (≥4 food groups). *** Odds Ratio and 95% Confidence Interval; † Reference group.

**Table 3 ijerph-20-00932-t003:** Multivariate analysis of factors associated with Khmer household dietary diversity in Tri Ton district.

Variables	aOR (95% CI) *	*p*-Value
Sex of household head		
Female ^†^	1	
Male	3.34 (1.86–5.98)	0.000
Education level of household head		
Illiteracy (Vietnamese) ^†^	1	
Literacy (Vietnamese)	9.32 (5.06–17.16)	0.000
Household income (million Vietnamese Dong/month)		
≤2.0 ^†^	1	
2.1–4.0	2.87 (1.56–5.29)	0.001
4.1–6.0	3.64 (1.03–12.93)	0.046
>6.0	6.27 (1.09–36.06)	0.040
Media exposure on nutrition and health		
No ^†^	1	
Yes	1.89 (1.04–3.41)	0.036
Frequency of eating (per day)		
Two times ^†^	1	
Three times and above	2.12 (1.13–3.98)	0.019

* Adjusted Odds Ratio and 95% Confidence Interval; † Reference group.

## Data Availability

The data presented in this study are available upon reasonable request from the corresponding author. The data are not publicly available due to the privacy of ethnic minority group.

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
