# Peer review of "Household Dietary Diversity among the Ethnic Minority Groups in the Mekong Delta: Evidence for the Development of Public Health and Nutrition Policy in Vietnam"

_ijerph, 2023, doi:10.3390/ijerph20020932_

Round 1
Reviewer 1 Report
Dear Authrors,
I have some recommendations to improve the quality of your paper. The scientific gap shuold be formulated at the end of the introduction part.
The sampling method and procedure are difficult to understand. It is hard to get a good grip on the whole procedure, so a table or figure would be very useful.
This paper has policy implications. This impact is very general, but it should be more detailed. The changes in food assistance programs or nutrition programs should be mentioned. So the required modifications or improvement of the recent measures. It woulg be also intresting if you could mention in the introduction part the main policy measures applied recently. In this way the scientific gap or impact can be highlighted.
At the limitation part you should mention only the limitations, the strenghts should be reformulated and used as a scientific impact.
I think it is also necessary to consider that the gender of the head of the family is related to the level of earnings and through this to the financial situation of the family.
The formal requirements should be checked as they are not appropriate.
Author Response
Response to Reviewer 1 Comments
Thank you very much for your reviews and valuable comments about the article.
Most of the suggestions were followed up and the article improved.
- The scientific gap shuold be formulated at the end of the introduction part.
Thank you for your comment. We consider your feedback and provide a sentence indicating the scientific gap (line 103 - 106)
- The sampling method and procedure are difficult to understand. It is hard to get a good grip on the whole procedure, so a table or figure would be very useful.
Thank you for your comment. We have taken your comment into account and discussed carefully with co-authors. Our study was designed a a cross-sectional study. The selection of the district, communes and households was performed using the random sampling technique. Therefore, we think that what we already provided from the sampling method and procedure (line 130 - 135) is sufficient and an added table or figure may repeat the information. We also note that the second reviewer general comments where they note that, “The authors have clearly answered the research question, the methodology appears to be sound and the results have already been incorporated into the new National Nutrition Strategy”
- This paper has policy implications. This impact is very general, but it should be more detailed. The changes in food assistance programs or nutrition programs should be mentioned. So the required modifications or improvement of the recent measures. It woulg be also intresting if you could mention in the introduction part the main policy measures applied recently. In this way the scientific gap or impact can be highlighted.
Thank you for your comment. We have revised the introduction section accordingly to your comment (line 75 - 80). In addition, the conclusion section was also revised. The policy impact was detailed and written “Prospective policy targeted childhood malnutrition control should be issued in consideration of diversity of culture, gender impact, and appropriate education and communication channels” (line 380 - 382).
- At the limitation part you should mention only the limitations, the strenghts should be reformulated and used as a scientific impact.
Thank you for your comment. We moved the strength part to the end of the discussion section (line 361 - 366)
- I think it is also necessary to consider that the gender of the head of the family is related to the level of earnings and through this to the financial situation of the family.
Thank you for your comment. We have added a sentence “which is opposite to male as household’s head. Additionally, income of male may be higher than females” along with a reference (line 322 - 323).
- The formal requirements should be checked as they are not appropriate.
Thank you for your comment. We have had a support from a native English speaker’s colleague to check and guarantee for academic English.
Reviewer 2 Report
Summary
Thank you for giving me the opportunity to review the manuscript. The aims of the study are to measure the level of dietary diversity among the ethnic Khmer minorities in Southwest Vietnam and identify factors that can be used for developing intervention strategies aimed at increasing food diversity and better nutrition among the ethnic minority groups in Vietnam. Although ethnic disparities in nutrition health (especially among those of minority status) are well known, contributing factors for these disparities are underexplored. In this sense, the current study makes an important contribution especially for policymaking aimed at reducing nutrition disparities, which is a highly relevant public and global health issue.
General comments
The authors have clearly answered the research question, the methodology appears to be sound and the results have already been incorporated into the new National Nutrition Strategy. The manuscript is well structured and well written. These are the strengths. The major shortcoming of the study is that the result does not provide evidences on characteristics specific to the minority Khmer populations in Vietnam and thus what special needs they have. Most factors associated with dietary diversity identified in the study are already well-known, and the authors could elaborate (perhaps in the discussion section) how these factors impact the dietary patterns of the minority ethnic populations different from those of the majority populations. The paper will substantially improve in quality if such comparison could be made.
The authors could consider replacing the term “developing countries” with “low-income countries” or “middle-income countries” as reported in the original UNICEF article which the authors cite. The term “developing countries”, are mentioned several times in the manuscript, and are not appropriate for a global health article. Also see https://gh.bmj.com/content/7/6/e009704.
Line 261: The abbreviation HDDS should be spelled out when it first appears in text (not in line 261).
Specific comments
Introduction
The introduction provides a good overview. I recommend the authors to elaborate in what ways socioeconomic and cultural differences influence nutrition disparities among the Khmer populations by referring to relevant articles (ex. low socioeconomic status held by majority of Khmer populations leading to limited income generating opportunities which leads to not being able to buy fruit or meat; concentrated in fishery industry; low literacy level due to language issues and thus low integration in the Vietnamese mainstream education system – leading to low health literacy etc.,).
Here, it might be worthwhile to mention some political, historical and geographical contexts too. Books such as the following could be relevant: "Persistent Malnutrition in Ethnic Minority Communities of Vietnam: Issues and Options for Policy and Interventions" https://doi.org/10.1596/978-1-4648-1432-7
If possible, include data or literature that compares the nutrition status of the Khmer ethnic group (33%) with that of other dominant ethnic groups (67%) within the Tri Ton district (in addition to the comparison with the national average). The current text seems to indicate that the entire Tri Ton district is affected by external factors such as climate change – which may have equally negative impact on the majority ethnic populations too. Even if this is so, the authors should make an argument that the minority populations are the worse effected in the local context of Tri Ton district, as this is the main focus point of the article. This justifies the study aim of investigating into the factors that are especially relevant for the ethnic minority group and informing the National Nutrition Strategy that have specific focus on the ethnic minority populations.
Line 78: English (“high risk groups” or “at high risks”)
Methods
Study design, settings and participants
Line 111-113: The background information on the recent young Khmer migrants to the urban cities is interesting, but what is the relevance? Recommendation: elaborate here or revisit this topic in the Discussion section.
In addition, an overview of socioeconomic and demographic data/information on the Khmer population living in the Tri Ton district will be very relevant and informative here. For example, are they concentrated in a specific sector that requires manual labour, fisheries or agriculture, compared to the dominant ethnic populations? What about their literacy level or school-completion rates? Since these are asked in the survey, the general statistical information here (especially in comparison to the majority ethnic group) will be meaningful and could be revisited in the Discussion section again.
I am not an expert on statistical methods and I would like to ask other reviewers to assess their quality, but the procedures appear to be sound.
Results
The results section should solely present the findings from the conducted survey, without making comparison with the national average or other statistical data. Or the authors need to do this in a very systematic way. For some features, comparisons are made (ex. income) and for some, not (ex. farm land ownerships). This is rather confusing and not consistent.
Option one: Only present results from the survey. Comparison with other data can be made in the Discussion Section systematically to highlight the specific features of the populations in concern.
Option two: Clearly state in the Introduction that making comparison with existing data to highlight features characteristics to the Khmer ethnic minority populations in the Tri Ton District is one of the study aims. In this way, it is justifiable to present the comparison in the Result section (but then it has to be done for all variables).
Line 196: Authors need to clarify “higher” than what? National average? As the percentage of female-headed households is 40%, it is not “higher” than male-headed households. If this is “higher” than the Tri Ton average or national average, then this is an interesting finding and the implication could be discussed in the Discussion section.
Line 201-202: consider different English expression.
Discussion
As mentioned before, I would recommend to elaborate on the comparison with other existing data and studies. The authors need to clearly separate: 1) the specific features and factors relevant to the Khmer ethnic group in concern, and 2) what could be relevant to nutrition strategy in rural context of Vietnam in general.
Line 261-264: In what way is the study finding consistent with reference 48? Here the comparison with the national average in Vietnam would be very important and interesting. It should be discussed separately with the comparison with Cambodia [reference 49] where 97.6% are Khmer. Please clarify, the arguments you are trying to make.
Line 263-264: Similarly, I recommend the authors to make it clear what their major argument is by comparing the result with a very different context in Mexico: ex. justify the comparison by clarifying that the study in Mexico was also done among minority populations in a rural context.
Line 265-266: The current study does not provide evidence that the DD is lower among the ethnic Khmer compared to the general population. If the authors would like to make this point, a clearer comparison with a national data or other studies need to be made.
Line 277-290: The purpose of the arguments made here are rather unclear. Is this an argument for overcoming the limitation of only asking the survey question once in the last 24 hours? (Then it should be part of the Limitation section). Or did the authors want to argue that long held habits are difficult to change? But are not these habits the results of the low-socio economic status of the population in concern, which has been explored in the survey?
Line 291-294: Socioeconomic status influencing the HDDS was one of the findings of the study, so it does not make sense to list it here.
Line 297-348: “Sex and educational level of the household head, household income, media exposure to nutrition and health, and frequency of eating were significantly associated with diversified diets” – these findings do not appear to be specific to the ethnic minority population but also apply to the general Vietnamese population.
All other literature listed also seem to confirm the general situation in other low to middle-income countries and not specific to minority ethnic populations nor ethnic minorities in Vietnam. In my opinion, the whole manuscript will be significantly improved if the authors elaborate here how the minority populations are particularly affected by these factors that affect food diversity and nutrition related health outcomes. The comparisons with other very different contexts in different countries need to be done with care and with justification.
Limitations and strength
I suggest bringing this section right after Discussion or part of the Discussion section (see my comments in Discussion section).
I would also recommend including, as one of the limitations, that comparison with non-minority populations living in the study area could not be made. Such comparison would have had made it clear what the general problems are, and what are the issues and needs specific to ethnic minority populations in Vietnam. To overcome this, comparison can be made with existing data available in Vietnam, which, as I suggested before, could be elaborated in the discussion section.
Conclusion
I think this part is clear. I recommend, however, to bring it to the end.
Author Response
Response to Reviewer 2 Comments
Thank you very much for your reviews and valuable comments about the article.
Most of the suggestions were followed up and the article improved.
- Summary
Thank you for giving me the opportunity to review the manuscript. The aims of the study are to measure the level of dietary diversity among the ethnic Khmer minorities in Southwest Vietnam and identify factors that can be used for developing intervention strategies aimed at increasing food diversity and better nutrition among the ethnic minority groups in Vietnam. Although ethnic disparities in nutrition health (especially among those of minority status) are well known, contributing factors for these disparities are underexplored. In this sense, the current study makes an important contribution especially for policymaking aimed at reducing nutrition disparities, which is a highly relevant public and global health issue.
Thank you very much for your comment. We appreciate your positive feedback.
- General comments
The authors have clearly answered the research question, the methodology appears to be sound and the results have already been incorporated into the new National Nutrition Strategy. The manuscript is well structured and well written. These are the strengths. The major shortcoming of the study is that the result does not provide evidences on characteristics specific to the minority Khmer populations in Vietnam and thus what special needs they have. Most factors associated with dietary diversity identified in the study are already well-known, and the authors could elaborate (perhaps in the discussion section) how these factors impact the dietary patterns of the minority ethnic populations different from those of the majority populations. The paper will substantially improve in quality if such comparison could be made.
Thank you very much for your comment. We compared the dietary pattern of Khmer group with the majority populations in Vietnam (line 276 - 278). However, we have insufficient data to discuss further the impacts of associated factors.
- The authors could consider replacing the term “developing countries” with “low-income countries” or “middle-income countries” as reported in the original UNICEF article which the authors cite. The term “developing countries”, are mentioned several times in the manuscript, and are not appropriate for a global health article. Also see https://gh.bmj.com/content/7/6/e009704.
Thank you very much for your comment. We replaced all the term “developing countries” by “low – and middle-income countries”.
- Line 261: The abbreviation HDDS should be spelled out when it first appears in text (not in line 261).
Thank you very much for your comment. We revised your comment.
Specific comments
- Introduction
The introduction provides a good overview. I recommend the authors to elaborate in what ways socioeconomic and cultural differences influence nutrition disparities among the Khmer populations by referring to relevant articles (ex. low socioeconomic status held by majority of Khmer populations leading to limited income generating opportunities which leads to not being able to buy fruit or meat; concentrated in fishery industry; low literacy level due to language issues and thus low integration in the Vietnamese mainstream education system – leading to low health literacy etc.,).
Thank you very much for your comment. We elaborated the ideas as you suggested. We borrowed your ideas to develop the sentences (line 83 - 87).
- Here, it might be worthwhile to mention some political, historical and geographical contexts too. Books such as the following could be relevant: "Persistent Malnutrition in Ethnic Minority Communities of Vietnam: Issues and Options for Policy and Interventions" https://doi.org/10.1596/978-1-4648-1432-7
Thank you very much for your comment and the reference. We went through the reference and revised the introduction section as per your advice. The added sentences were written: “Several policies and programs, along with significant investments, have been allocated to vulnerable communities such as the Khmer minority group to improve livelihood and health, and nutrition status. The National Nutrition Strategy and the targeted national program for socio-economic development in ethnic minorities and mountainous areas, which have been issued recently, also created a supportive environment for the comprehensive development of ethnic people.” (line 75 - 80)
- If possible, include data or literature that compares the nutrition status of the Khmer ethnic group (33%) with that of other dominant ethnic groups (67%) within the Tri Ton district (in addition to the comparison with the national average). The current text seems to indicate that the entire Tri Ton district is affected by external factors such as climate change – which may have equally negative impact on the majority ethnic populations too. Even if this is so, the authors should make an argument that the minority populations are the worse effected in the local context of Tri Ton district, as this is the main focus point of the article. This justifies the study aim of investigating into the factors that are especially relevant for the ethnic minority group and informing the National Nutrition Strategy that have specific focus on the ethnic minority populations.
Thank you very much for your comment. Our literature review could not find any data specifically for nutrition status of people in Tri Ton district by dominant ethnic groups. However, we revised the introduction section in considering the impacts of climate change on the Khmer group. (line 93 - 98).
- Line 78: English (“high risk groups” or “at high risks”)
Thank you very much for your comment. We corrected the term with “at high risks” (line 70).
- Methods
Study design, settings and participants
Line 111-113: The background information on the recent young Khmer migrants to the urban cities is interesting, but what is the relevance? Recommendation: elaborate here or revisit this topic in the Discussion section.
Thank you very much for your comment. We added some sentences to elaborate the ideas “Moreover, while young Khmer people were in the cities and industrial zones, the older people remained at home to take care of the children and prepared meals for the family. The traditional custom in eating habits from the older generation, such as the predominantly starchy and vegetables food based diet, was persistent in the households” in the Discussion section (line 281 - 285).
- In addition, an overview of socioeconomic and demographic data/information on the Khmer population living in the Tri Ton district will be very relevant and informative here. For example, are they concentrated in a specific sector that requires manual labour, fisheries or agriculture, compared to the dominant ethnic populations? What about their literacy level or school-completion rates? Since these are asked in the survey, the general statistical information here (especially in comparison to the majority ethnic group) will be meaningful and could be revisited in the Discussion section again.
Thank you very much for your comment. We took your ideas into account and searched for further information. The people in Tri Ton district have a low education level and their livelihood mainly depends on the forest and small-scale cultivation, which lead them being more vulnerable with climate change and malnutrition. We revised the text “The most recent data showed that people in 14/15 communes in the district completed the primary school literacy. Livelihood of local people heavily depend on forest and small-scale cultivation” (line 91 - 93)
- I am not an expert on statistical methods and I would like to ask other reviewers to assess their quality, but the procedures appear to be sound.
Thank you very much for your comment. We appreciate your feedback. We already carefully with statistical methods and consulted with our statistician colleague.
- Results
The results section should solely present the findings from the conducted survey, without making comparison with the national average or other statistical data. Or the authors need to do this in a very systematic way. For some features, comparisons are made (ex. income) and for some, not (ex. farm land ownerships). This is rather confusing and not consistent. Option one: Only present results from the survey. Comparison with other data can be made in the Discussion Section systematically to highlight the specific features of the populations in concern. Option two: Clearly state in the Introduction that making comparison with existing data to highlight features characteristics to the Khmer ethnic minority populations in the Tri Ton District is one of the study aims. In this way, it is justifiable to present the comparison in the Result section (but then it has to be done for all variables).
Thank you very much for your comment. We revised the content corresponding to your option one (line 199 - 200)
- Line 196: Authors need to clarify “higher” than what? National average? As the percentage of female-headed households is 40%, it is not “higher” than male-headed households. If this is “higher” than the Tri Ton average or national average, then this is an interesting finding and the implication could be discussed in the Discussion section.
Thank you very much for your comment. We revised and made the data more clearer. The sentence was rewrited as “As presented in Table 1, the proportion of female household heads, illiteracy (Vietnamese), and age range from 41 to 65 years were 40.3%, 21.9%, and 54.5%, respectively” (line 197 - 198).
- Line 201-202: consider different English expression.
Thank you very much for your comment. We revised the sentences to “The mean household dietary diversity score was 4.6 (SD ±1.4). The majority of the households were at a moderate level of dietary diversity (over 70%). The proportion of households classified as having diversified dietary intake was roughly 80%”. (line 203 - 205).
- Discussion
As mentioned before, I would recommend to elaborate on the comparison with other existing data and studies. The authors need to clearly separate: 1) the specific features and factors relevant to the Khmer ethnic group in concern, and 2) what could be relevant to nutrition strategy in rural context of Vietnam in general.
Thank you very much for your comment.
- We provided more details related to associated factors to the Khmer ethnic group “This could possibly mean that females (who in many cultures are food makers) [66] did not have sufficient time and resources required to provide optimum dietary diversity for their households [67-69], which is opposite to males as the household’s head. Additionally, the income of males may be higher than females” (line 320 - 323).
We also added some information in the introduction section regarding to this request.
- We provided information as advised “The study has already made significant contributions to the body of knowledge, including informing the development of the National Nutrition Strategy for the period of 2021-2030, envisioned to be revised in 2045. From our understanding, this is the first research on the Khmer minority group, particularly for household dietary diversity. Results from this study will raise awareness among the community and policymakers regarding improving household dietary diversity to control undernutrition” (line 361 - 366).
- Line 261-264: In what way is the study finding consistent with reference 48? Here the comparison with the national average in Vietnam would be very important and interesting. It should be discussed separately with the comparison with Cambodia [reference 49] where 97.6% are Khmer. Please clarify, the arguments you are trying to make.
Thank you very much for your comment. We rechecked the reference 48 and 49, and made revision corresponding to your advice. “The mean household dietary diversity score (HDDS) of this study was consistent with the previous studies in Cambodia [49], where 97.6% of the population is ethnically Khmer, and greater than 80% of them completed primary school education [50]. In contrast, the mean HDDS from our study was lower than those in a survey conducted in four provinces in Vietnam in 2011, where the Khmer group was rarely found” (line 262 - 266).
- Line 263-264: Similarly, I recommend the authors to make it clear what their major argument is by comparing the result with a very different context in Mexico: ex. justify the comparison by clarifying that the study in Mexico was also done among minority populations in a rural context.
Thank you very much for your comment. We realised that the comparison made between our study and the Mexico is not relevant. Therefore, we decided to move out the sentence for Mexico comparison.
We wanted to emphasize the importance of ethnicity in improving HDD, therefore we have revised the second paragraph in the Discussion section. The paragraph is written:
“The mean household dietary diversity score (HDDS) of this study was consistent with the previous studies in Vietnam [48] and Cambodia [49], where 97.6% of the population is ethnically Khmer, and greater than 80% of them completed primary school education [50]. In contrast, the mean HDDS from our study was lower than those in a survey conducted in four provinces in Vietnam in 2011 where the Khmer group was rarely found (Ref 48). A number of studies in low- and middle-income nations also found the similarity that most ethnic groups had low or medium dietary diversity scores [37, 52] [53, 54]. Thus, ethnicity may have impacts on HDDS as people in the same ethnic group are likely to have similar dietary patterns and eating behaviours independent of socioeconomic status. However, we have a limitation to in-depth investigation for this statement. In addition, a considerable proportion of households with Low Dietary Diversity (consumed less than three food groups per day) in this study may result in increased malnutrition in the households [16, 55]. Future studies regarding HDD should consider ethnicity in order to control malnutrition” (line 262 - 275).
- Line 265-266: The current study does not provide evidence that the DD is lower among the ethnic Khmer compared to the general population. If the authors would like to make this point, a clearer comparison with a national data or other studies need to be made.
Thank you very much for your comment. This comment was explained in the paragraph answered for Comment number 17 (above).
- Line 277-290: The purpose of the arguments made here are rather unclear. Is this an argument for overcoming the limitation of only asking the survey question once in the last 24 hours? (Then it should be part of the Limitation section). Or did the authors want to argue that long held habits are difficult to change? But are not these habits the results of the low-socio economic status of the population in concern, which has been explored in the survey?
Thank you very much for your comment. We have provided a discussion of 24 hours recall limitation in the Limitation study section. Because our study was a cross-sectional study, we do not have a power to define a causual relationship between HDD and low socioeconomic status. Therefore, we revised the sentences to make them more clearer.
“While we acknowledge that practicing dietary diversity is important, it is worth noting that improved dietary diversity can be challenging due to a range of factors including, socio-economic, residence, and cultural factors [62, 63], especially where traditional food menus with low diversity are favoured” (line 297 – 301).
- Line 291-294: Socioeconomic status influencing the HDDS was one of the findings of the study, so it does not make sense to list it here.
Thank you very much for your comment. We took out the “socioeconomic status” phrase (line 302).
- Line 297-348: “Sex and educational level of the household head, household income, media exposure to nutrition and health, and frequency of eating were significantly associated with diversified diets” – these findings do not appear to be specific to the ethnic minority population but also apply to the general Vietnamese population.
Thank you very much for your comment. We added information in the lines mentioned “Consistent with the literature review for general populations, our final logistic regression model revealed that factors such as the sex and educational level of the household head, household income, media exposure to nutrition and health, and frequency of eating were significantly associated with diversified diets in the Khmer group” (line 307 - 310).
- All other literature listed also seem to confirm the general situation in other low to middle-income countries and not specific to minority ethnic populations nor ethnic minorities in Vietnam. In my opinion, the whole manuscript will be significantly improved if the authors elaborate here how the minority populations are particularly affected by these factors that affect food diversity and nutrition related health outcomes. The comparisons with other very different contexts in different countries need to be done with care and with justification.
Thank you very much for your comment. We appreciate your suggestions. However, due to the lack of evidence for the minority groups, particularly HDD data in ethnic populations, we will investigate in the future studies.
- Limitations and strength
I suggest bringing this section right after Discussion or part of the Discussion section (see my comments in Discussion section).
Thank you very much for your comment. We moved the strength to the end of the Discussion section (line 361 – 366).
- I would also recommend including, as one of the limitations, that comparison with non-minority populations living in the study area could not be made. Such comparison would have had made it clear what the general problems are, and what are the issues and needs specific to ethnic minority populations in Vietnam. To overcome this, comparison can be made with existing data available in Vietnam, which, as I suggested before, could be elaborated in the discussion section.
Thank you very much for your comment. We added a sentence indicating the limitation you mentioned “We also faced a shortfall in comparing the findings with the Kinh group in the district” (line 392 - 393).
- Conclusion
I think this part is clear. I recommend, however, to bring it to the end.
Thank you very much for your comment.